# Tamoxifen Activates Transcription Factor EB and Triggers Protective Autophagy in Breast Cancer Cells by Inducing Lysosomal Calcium Release: A Gateway to the Onset of Endocrine Resistance

**DOI:** 10.3390/ijms25010458

**Published:** 2023-12-29

**Authors:** Cecilia Boretto, Chiara Actis, Pawan Faris, Francesca Cordero, Marco Beccuti, Giulio Ferrero, Giuliana Muzio, Francesco Moccia, Riccardo Autelli

**Affiliations:** 1Department of Clinical and Biological Sciences, University of Turin, 10125 Turin, Italy; cecilia.boretto@unito.it (C.B.); chiara.actis@unito.it (C.A.); giulio.ferrero@unito.it (G.F.); giuliana.muzio@unito.it (G.M.); 2Department of Brain and Behavioral Sciences, University of Pavia, 27100 Pavia, Italy; pawan.faris@unipv.it; 3Department of Computer Science, University of Turin, 10149 Turin, Italy; francesca.cordero@unito.it (F.C.); marco.beccuti@unito.it (M.B.); 4Laboratory of General Physiology, Department of Biology and Biotechnology “L. Spallanzani”, University of Pavia, 27100 Pavia, Italy; francesco.moccia@unipv.it

**Keywords:** breast cancer, endocrine resistance, lysosomal calcium channels, tamoxifen, TFEB, calcium signaling, TRPML1, TPCs

## Abstract

Among the several mechanisms accounting for endocrine resistance in breast cancer, autophagy has emerged as an important player. Previous reports have evidenced that tamoxifen (Tam) induces autophagy and activates transcription factor EB (TFEB), which regulates the expression of genes controlling autophagy and lysosomal biogenesis. However, the mechanisms by which this occurs have not been elucidated as yet. This investigation aims at dissecting how TFEB is activated and contributes to Tam resistance in luminal A breast cancer cells. TFEB was overexpressed and prominently nuclear in Tam-resistant MCF7 cells (MCF7-TamR) compared with their parental counterpart, and this was not dependent on alterations of its nucleo-cytoplasmic shuttling. Tam promoted the release of lysosomal Ca^2+^ through the major transient receptor potential cation channel mucolipin subfamily member 1 (TRPML1) and two-pore channels (TPCs), which caused the nuclear translocation and activation of TFEB. Consistently, inhibiting lysosomal calcium release restored the susceptibility of MCF7-TamR cells to Tam. Our findings demonstrate that Tam drives the nuclear relocation and transcriptional activation of TFEB by triggering the release of Ca^2+^ from the acidic compartment, and they suggest that lysosomal Ca^2+^ channels may represent new druggable targets to counteract the onset of autophagy-mediated endocrine resistance in luminal A breast cancer cells.

## 1. Introduction

Breast cancer is the most prevalent malignancy in women worldwide; the World Health Organization has reported that in 2020 about 2.3 million new cases were diagnosed and about 690,000 deaths recorded [1]. Breast cancer presents as a highly heterogeneous disease, so that it is currently classified in up to five surrogate intrinsic subtypes [2]. On the whole, the luminal A-like is the most frequent subtype, accounting for about 70% of total diagnoses, and the one with the most favorable prognosis [3]. Tamoxifen (Tam) in either neoadjuvant or adjuvant settings still represents the most common and effective treatment for this breast cancer subtype [4]. Despite the effectiveness of the current therapeutic protocols, patient survival is impaired by the onset of resistance to Tam or to other antiestrogen drugs, a condition defined as endocrine resistance [5]. The latter may result either from intrinsic refractoriness to therapy (‘de novo’ endocrine resistance) or from a slow adaptation process that over time renders breast cancer cells unresponsive to the endocrine drugs (‘acquired’ resistance) [6,7].

Resistance to anticancer drugs relies on several mechanisms: among them, activation of macroautophagy (from hereafter autophagy) has emerged as a key factor. In the last years, autophagy has been revealed to play a dual role in cancer [8]. While functional autophagy effectively prevents neoplastic transformation, in established cancers activated autophagy represents a factor favoring tumor cell survival and the onset of drug resistance [9]. We have previously demonstrated that autophagic flux is increased in a subline of MCF7 cells resistant to Tam (MCF7-TamR) and contributes to their drug resistance [10]. Other reports have indeed demonstrated that Tam actually triggers autophagy in target cells [10,11,12] and promotes the nuclear localization and activation of transcription factor EB (TFEB) [13], the master regulator of autophagy [14]. Collectively, these findings support the view that anticancer drugs may partake in positively regulating autophagy in cancer cells. However, the fine molecular mechanisms by which Tam, as well as other anticancer drugs, modulate TFEB activity and, in turn, autophagic flux are so far mostly unclarified.

TFEB, together with TFE3, TFEC and MITF, belongs to the microphthalmia family of transcription factors [14,15] and stimulates the expression of genes involved in lysosome biogenesis and autophagy by binding the coordinated lysosomal expression and regulation consensus motif present in the promoter of these genes [14,16]. The transcriptional activity of TFEB/TFE3 is strictly controlled by serine phosphorylation due to various kinases, including mTORC1 and ERK2/MAPK1 [14,17,18,19], which induce the reversible interaction of TFEB/TFE3 with the chaperone 14-3-3 and their cytoplasmic retention in an inactive state. Nutrient starvation or lysosomal stress lead to the calcineurin-mediated dephosphorylation of the transcription factors and their dissociation from 14-3-3. These events cause the nuclear relocation of both TFEB and TFE3, and the ensuing transcriptional activation of their target genes [20].

Transient receptor potential mucolipin 1 (TRPML1, also known as MCOLN1) is a lysosomal Ca^2+^-releasing channel that is activated by nutrient starvation and thereby stimulates the Ca^2+^/calmodulin-dependent calcineurin to dephosphorylate TFEB, eventually promoting its nuclear translocation [20,21,22]. In addition, the nuclear translocation of TFEB can be induced by activation of the two-pore channels (TPCs) [23], which represent an additional pathway for lysosomal Ca^2+^ mobilization [22]. A recent study, however, showed that inositol-1,4,5-trisphosphate (InsP_3_) can also activate TFEB and induce lysosome biogenesis by determining Ca^2+^ release from the endoplasmic reticulum (ER) [24], the most abundant intracellular Ca^2+^ reservoir. Previous reports have evidenced that Tam increases the intracellular Ca^2+^ concentration ([Ca^2+^]_i_) in MCF7 cells through the G protein-coupled estrogen receptor 1 (GPER1), although the underlying signaling machinery is still unknown [25,26]. Similarly, it is unknown whether Tam promotes the calcineurin-mediated dephosphorylation and transcriptional activation of TFEB through an increase in [Ca^2+^]_i_.

Given the relevance of autophagy activation in endocrine resistance, understanding how Tam can modulate the autophagic flux becomes of paramount importance to restrain or revert Tam resistance in breast cancer cells. Therefore, in the present research we have investigated and characterized the fine molecular mechanisms by which the transcriptional activity and subcellular localization of TFEB are modulated by Tam and contribute to Tam resistance. Our results demonstrate that Tam affects [Ca^2+^]_i_, which in turn accounts for the observed nuclear relocation and transcriptional activation of TFEB, and they suggest that interfering with the Tam-induced alteration of Ca^2+^ homeostasis restores the susceptibility of MCF7-TamR cells to the effect of the drug.

## 2. Results

### 2.1. TFEB and Some of Its Target Genes Are Overexpressed in MCF7-TamR Cells

We have previously demonstrated that MCF7-TamR cells have a greater autophagic flux than the parental cell line MCF7, and that pharmacological inhibition of autophagy partly restores their susceptibility to Tam [10]. Since the expression of most of the autophagy- and lysosome-related genes is under the transcriptional control of TFEB [14], we focused on dissecting the role of this transcription factor and of its target genes in contributing to the resistance of MCF7-TamR cells. RNA-sequencing (RNA-seq) data generated by the analysis of the transcriptomes of MCF7 and MCF7-TamR cells revealed that TFEB was significantly upregulated in MCF7-TamR cells (Figure 1a; Log2 Fold Change was 0.69 and the Adj *p* value = 6.19 × 10^−4^). This result was confirmed by semiquantitative Real-Time RT-qPCR and Western blotting (Figure 1b,c).

As expected, RNA-seq revealed that several of the TFEB target genes were also upregulated in MCF7-TamR cells (Figure 1a). Among the most upregulated genes, those encoding for some subunits of the vATPase (ATP6V1A, ATP6V1C1, ATP6V1E1, ATP6V1D), cathepsins (cathepsin B, F, and A), and other lysosomal proteins were included. Interestingly, TRPML1, a lysosomal Ca^2+^ channel previously reported to be critical for the activation of both TFEB and autophagy [20,27], was also significantly upregulated in MCF7-TamR cells (Figure 1a). The RNA-seq data of some validated TFEB targets impacting autophagy [28] were further validated by Real-Time RT-qPCR (Figure 1d).

The observed upregulation of TFEB and of a subset of its target genes in MCF7-TamR cells indicates that both the amount and the biological activity of this transcription factor are increased, supporting our previous finding of a greater autophagic flux in these cells compared with the parental ones [10].

### 2.2. Subcellular Localization of TFEB in MCF7 and MCF7-TamR Cells

The transcriptional activity of TFEB requires its translocation from the cytoplasm to the nucleus [14]. We next investigated whether the upregulation of TFEB target genes detected in MCF7-TamR cells was accounted for by its increased nuclear localization. To this end, immunofluorescence was used to investigate the localization of endogenous TFEB, and these results were compared with those produced by either transient or stable transfection of a TFEB-GFP chimera (Figure 2a,b, respectively).

Confocal microscopy revealed that the fraction of cells with nuclear TFEB was significantly greater in MCF7-TamR cells compared to their parental counterpart, and that this result was independent of the technique used (Figure 2c,d). However, transient transfection of TFEB-GFP produced the greatest difference of fluorescence intensity between the nuclei without or with TFEB inside (Figure 2e, blue or orange dots, respectively), which conferred to this method a specificity and sensitivity higher than those of immunofluorescence. Similar results were also obtained in experiments performed with cells stably transfected with TFEB-GFP (Figure 2f). On the basis of our results and those reported by other research groups [18,20,29], either transient or stable TFEB-GFP transfectants were then used in the subsequent experiments to investigate the subcellular localization of TFEB.

Our findings thus definitely confirm that TFEB is prominently nuclear in MCF7-TamR cells and agree with the increased transcriptional activity of this transcription factor in Tam-resistant cells compared with the parental ones.

To go more deeply into the causes of the high nuclear relocation of TFEB in MCF7-TamR cells, we focused on the molecular mechanisms controlling its subcellular localization. At first, we verified whether the physiological processes governing the nucleo-cytoplasmic shuttling of TFEB were normally operating in MCF7 cells. To this aim, we tested the effect of starvation, a stimulus well-known to drive the nuclear relocation of TFEB. MCF7 cells were starved for 4 h in Hank’s balanced salt solution (HBSS), which effectively led to the nuclear translocation of the transcription factor. Refeeding for 2 h with normal growth medium fully reversed the starvation-induced nuclear translocation of TFEB, and this effect was completely abrogated by leptomycin B (LMB), an inhibitor of nuclear export (Appendix A).

Subsequently, we investigated whether Tam affects the subcellular distribution of TFEB in these cells also, in addition to those previously reported [13]. MCF7 cells stably expressing TFEB-GFP were treated or not with 5 µM Tam for 24 h: confocal analysis showed that the drug determined the nuclear translocation of TFEB in a large number of cells (Appendix A). The effect was almost completely reversed by removal of Tam from the culture medium for 24 h (washout), which determined the active efflux of the transcription factor from the nucleus and demonstrated that its Tam-induced nuclear relocation was reversible. The export of TFEB from the nucleus triggered by the Tam washout was totally abrogated when the washout was performed in the presence of LMB. These results strongly indicate that Tam is capable of reversibly affecting the subcellular localization of TFEB in MCF7 cells also.

Based on the above results, we then verified whether the increased nuclear accumulation of TFEB observed in MCF7-TamR cells relies on the presence of Tam in the growth medium or, rather, depends on the alteration of the molecular mechanisms controlling its subcellular localization. To this end, MCF7-TamR cells stably expressing TFEB-GFP were shifted to a growth medium devoid of Tam for 24 h (Tam washout) before assessing the subcellular localization of TFEB. As already evidenced for the parental cells, Tam removal reduced the fraction of MCF7-TamR cells displaying nuclear TFEB; this effect was caused by the active export of the transcription factor, as demonstrated by its persistent nuclear retention when Tam washout was performed in the presence of LMB (Figure 3a).

To definitely rule out any possibility that the mechanism controlling the subcellular relocation of TFEB was altered in MCF7-TamR, the cells were grown in the absence of Tam for 24 h to allow the complete export of TFEB-GFP from the nuclei, starved for 4 h, and subsequently refed either in the absence or presence of LMB. In agreement with the results gathered with the parental cell line, starvation caused the nuclear accumulation of TFEB, which was effectively reversed by refeeding, and LMB completely prevented the nuclear export of TFEB brought about by refeeding subsequent to starvation (Figure 3b).

The above results demonstrate that the mechanisms controlling the subcellular localization of TFEB are conserved in MCF7-TamR cells and that Tam triggers the reversible relocation of TFEB to the nucleus without affecting its nuclear efflux. Our results thus strongly support the view that the greater nuclear accumulation of TFEB observed in MCF7-TamR cells results from the active induction of cytoplasm-to-nucleus relocation of the transcription factor specifically afforded by Tam.

### 2.3. Tam Induces an Increase in [Ca^2+^]_i_ in MCF7 Cells

The subcellular localization of TFEB is controlled by even small or spatially localized increases in [Ca^2+^]_i_ [20,21,24]. Therefore, we next investigated whether and how Tam induces intracellular Ca^2+^ signals in MCF7 cells. Tam caused a rapid increase in [Ca^2+^]_i_ that then declined to a sustained plateau (Figure 4a). 

The Ca^2+^ response to Tam was inhibited by G15 (1 µM), a selective GPER1 inhibitor (Figure 4a) [25], and mimicked by G1 (1 µM) (Figure 4b), a commercial GPER1 agonist [25]. The statistical analysis of these findings has been reported in Figure 4c. The Ca^2+^ response to chemical stimulation in MCF7 cells is usually triggered by intracellular Ca^2+^ mobilization, which is responsible for the initial Ca^2+^ peak, and sustained over time by store-operated Ca^2+^ entry (SOCE) [30,31]. Consistently, Tam evoked a rapid Ca^2+^ transient in the absence of extracellular Ca^2+^ (0Ca^2+^) (Figure 4d), which reflects mobilization of the endogenous Ca^2+^ reservoir; whereas, restitution of extracellular Ca^2+^ to the perfusate induced a second increase in [Ca^2+^]_i_ (Figure 4d), which reflects extracellular Ca^2+^ entry. As described elsewhere [32,33], Tam was removed 100 sec before Ca^2+^ re-addition to the bath to prevent the activation of second messengers-operated channels and to assess whether previous depletion of the endogenous Ca^2+^ pool was indeed able to elicit Ca^2+^ entry through store-operated channels. In agreement with this hypothesis, blocking SOCE with the selective inhibitor BTP-2 [32,33] significantly reduced the amplitude of the initial Ca^2+^ peak and converted the biphasic Ca^2+^ response to Tam into a transient increase in [Ca^2+^]_i_ (Figure 4e,f).

To decipher the signaling pathway driving intracellular Ca^2+^ mobilization, we first focused on InsP_3_ receptors (InsP_3_Rs), which represent the main ER Ca^2+^-releasing channel in MCF7 cells [30,34,35]. Tam-induced intracellular Ca^2+^ mobilization was suppressed by depleting the ER Ca^2+^ store with cyclopiazonic acid (CPA; 30 µM) (Figure 5a), a selective inhibitor of SERCA activity [32], and largely attenuated by 2-aminoethoxydiphenyl borate (2-APB; 50 µM) (Figure 5b), which selectively blocks InsP_3_Rs under 0Ca^2+^ conditions [32,36]. 

The statistical analysis of these findings has been reported in Figure 5c. However, the Tam-induced nuclear translocation of TFEB suggested that the lysosomal Ca^2+^-releasing channels TRPML1 and TPCs could contribute to Tam-evoked intracellular Ca^2+^ mobilization. Consistently, Tam-induced intracellular Ca^2+^ mobilization was significantly attenuated by depleting the lysosomal Ca^2+^ store with bafilomycin A1 (1 µM) (Figure 5d) [22,37], by inhibiting TRPML1 with ML-SI3 (100 µM) [38,39] (Figure 5d) or by blocking TPCs with NED-19 (100 µM) [22,37] (Figure 5d). The statistical analysis of these findings has been reported in Figure 5e.

Overall, these data confirm that Tam evokes an increase in [Ca^2+^]_i_ in MCF7 cells and provide the first evidence that this Ca^2+^ signal is initiated by GPER1 activation followed by ER Ca^2+^ release through InsP_3_Rs and lysosomal Ca^2+^ mobilization through TRPML1. InsP_3_-induced ER Ca^2+^ release, in turn, causes the depletion of ER Ca^2+^ that leads to SOCE activation on the plasma membrane [40].

### 2.4. Resting Ca^2+^ Levels Are Larger in MCF7-TamR Cells

The evidence that Tam evokes an increase in [Ca^2+^]_i_ in MCF7 cells (Figure 4) and that TFEB was mainly localized in the nucleus of MCF7-TamR cells (Figure 2) suggested that resting [Ca^2+^]_i_ was significantly larger in MCF7-TamR cells due to GPER1 activation. In agreement with this hypothesis, the resting [Ca^2+^]_i_ was significantly (*p* < 0.05) more elevated in MCF7-TamR as compared to MCF7 cells (Figure 6a). 

However, the resting [Ca^2+^]_i_ returned to the baseline after Tam washout from the medium (Figure 6a) or in the presence of G15 (1 µM) to inhibit GPER1 (Figure 6a). To further support the notion that larger basal Ca^2+^ levels observed in MCF7-TamR cells are due to an ongoing Ca^2+^ response to Tam, we monitored the Ca^2+^ signals evoked by Tam (5 µM) in MCF7-TamR cells after a rapid (10 min) washout of the agonist. Tam was still able to evoke an increase in [Ca^2+^]_i_ (Figure 6c), but this signal was dramatically lower and slower as compared to that recorded in MCF7 cells (Figure 6b). The statistical analysis of these data has been shown in Figure 6d.

Overall, these data strongly suggest that the nuclear localization of TFEB in MCF7-TamR cells is driven by an increase in resting [Ca^2+^]_i_ because of continuous Tam stimulation.

### 2.5. Pharmacological Inhibition of Lysosomal Ca^2+^ Channels Affects Tam-Induced Nuclear Relocation of TFEB

To verify the impact of Tam-induced changes in [Ca^2+^]_i_ we evaluated whether the pharmacological blockade of the underlying Ca^2+^ signal affected the Tam-induced nuclear translocation of TFEB in MCF7-TamR cells. Although both 2-APB and BTP-2 were ineffective in preventing the Tam-induced nuclear relocation of TFEB, NED-19 and ML-SI3 completely suppressed it (Figure 7a,b).

Subsequently, we verified whether impairing the Tam-induced nuclear relocation of TFEB using the Ca^2+^ signaling blockers affects the viability of MCF7-TamR cells. To gain this information, the cells were treated for five days with the Ca^2+^ channel inhibitors in the presence of 5 µM Tam. Whilst 2-APB and BTP-2 did not produce any significant effect, both NED-19 and ML-SI3 markedly reduced the viability of MCF7-TamR cells (Figure 7c).

This finding further indicates that the Tam-induced release of Ca^2+^ from the lysosomal compartment concurs to the cytoplasm-to-nucleus translocation of TFEB and that interfering with this mechanism effectively restores Tam susceptibility.

### 2.6. Silencing of TFEB, Yet Not of TFE3, Partly Sensitizes MCF7-TamR Cells to Tam

To verify whether TFEB and TFE3 play a role in the Tam resistance of MCF7-TamR cells, we downregulated TFEB, TFE3 or both simultaneously, and evaluated the viability of MCF7-TamR cells in the presence of Tam. Both single and double silencing reduced the amount of the corresponding mRNAs (Appendix A). Of interest, the downregulation of TFEB, yet not of TFE3, partly but significantly reduced the viability of MCF7-TamR cells. According to this result, when both targets were simultaneously downregulated, the observed reduction of viability was reminiscent of that brought about by TFEB silencing alone (Appendix A).

Collectively, these data confirm that TFEB plays a major role in granting the survival of MCF7-TamR cells in the presence of Tam. Blunting Tam-induced changes of intracellular Ca^2+^ homeostasis and the ensuing nuclear relocation of TFEB effectively restores the susceptibility of MCF7-TamR cells to the anticancer activity of Tam. 

### 2.7. Generation and Growth Characteristics of Additional Tam-Resistant Luminal A Breast Cancer Cell Lines

To further investigate whether the mechanisms of TFEB regulation by Tam in MCF7 cells are shared by other luminal A breast cancer cell lines, we derived new Tam-resistant cell lines from MDA-MB-415, T47D and ZR-75-1 cells using the same protocol adopted for establishing the MCF7-TamR cells [10]. Similar to what was observed for MCF7-TamR cells, the new Tam-resistant cell lines were able to grow in the presence of 5 µM Tam with a rate similar to that of the parental ones (Appendix A).

### 2.8. Effect of Tam on the Subcellular Localization of TFEB in Parental and Tam-Resistant MDA-MB-415, T47D and ZR-75-1 Cells

To verify whether Tam affects the subcellular localization of TFEB in both parental and Tam-resistant MDA-MB-415, T47D and ZR-75-1 cells, the cells were transiently transfected with TFEB-GFP and grown in the absence of the drug for 72 h to get rid of any nuclear TFEB. After this time, the cells were exposed (the parental cells) or re-exposed (the Tam-resistant cells) to 5 µM Tam for an additional 24 h. The confocal microscopy analysis revealed that Tam determined the relocation of TFEB-GFP to the nucleus in both the parental and Tam-resistant cell lines (Figure 8a,b).

These data confirm that Tam affects the subcellular relocation of TFEB in all the breast cancer cell lines of the luminal A subtype tested, likely representing a common effect of the drug on these cells.

## 3. Discussion

The standard therapy for early and advanced luminal A and B breast cancer in premenopausal patients mostly relies on the administration of Tam, the most used ‘endocrine’ drug. This treatment, effective in a large fraction of patients, is hampered in about 30% of them who, over time, develop Tam resistance and undergo disease progression [2,6,41]. Among the mechanisms accounting for endocrine resistance, the activation of autophagy is presently recognized as one of the most important ones in breast (as well as in several other types of) cancers [42,43,44]. Several independent studies have proved that Tam (and also other commonly used anticancer agents) increases the autophagic flux in target cells [45,46], potentially favoring the onset of endocrine resistance. The lack of detailed information at a molecular level as to how protective autophagy is triggered by anticancer treatment(s) holds back the development of effective strategies to restrain the onset of resistance or to resensitize the cells to the anticancer drugs.

TFEB finely modulates the expression of genes involved in both autophagy and lysosome biogenesis, playing the role of the most important physiological modulator of autophagy [14,16,18,47]. Consequently, the elevation in autophagic flux observed in drug-resistant cancer cells might result from either an increased availability of TFEB itself or from its increased transcriptional activity. In keeping with the above notion, our RNA-seq results show that TFEB and several of its target genes are upregulated in MCF7-TamR cells. Transcriptional regulation of TFEB is complex and also includes the intriguing capability of the transcription factor to self-regulate its own expression. In fact, by binding the coordinated lysosomal expression and regulation motif [16] present in the intron 1 of its gene, TFEB contributes to dictating its intracellular amount [48]. Due to this sort of self-amplification loop, the increased nuclear localization of TFEB induced by Tam might thus represent a factor concurring to its overexpression in breast cancer cells in the presence of the drug. In addition to the self-regulation mechanism depicted above, another critical factor potentially contributing to regulate TFEB expression has been identified in the activity of the master regulator of mitochondrial proliferation, PGC1α. Although being itself a target of TFEB, PGC1α may also regulate TFEB expression by binding a specific region of TFEB promoter [49] and triggering a transcriptional mechanism which involves the activity of PPARα/Retinoid X Receptor [49]. However achieved, the increased expression or transcriptional activity of TFEB eventually leads to the elevation of its target genes and to a greater autophagic flux. This data agrees and explains our previous finding that Tam-resistant cells are characterized by an autophagic flux greater than that of parental cells [10]. Here, we also demonstrate that, in MCF7-TamR cells, TFEB exhibits a predominantly nuclear localization which fully accounts for both the upregulation of some of its target genes and the more elevated autophagic flux detected in these, compared with the parental cells. The subcellular localization of TFEB is controlled by the convergent regulation of the phosphorylation and dephosphorylation of critical serines. The reversible phosphorylation of serines 142 and 211 leads to the interaction of TFEB with the 14-3-3 proteins hiding the nuclear localization signal present between the residues 241 and 252 of the human protein [18], eventually preventing its nuclear relocation. By contrast, calcineurin-mediated serine dephosphorylation totally reverts this process, leading to the nuclear relocation of TFEB. This complex interplay of post-translational modifications determines whether the transcription factor is blocked inside the cytoplasm in an inactive form or can translocate to the nucleus to drive the transcription of its genomic targets [14]. Besides alterations in phosphorylation status, changes in the subcellular localization of TFEB could be also ascribed to the dysregulation of its nuclear import or export. However, our results demonstrate that the mechanisms governing the subcellular localization of TFEB in response to starvation are conserved and normally operating in both MCF7 and MCF7-TamR cells. In fact, starvation determined the rapid and effective accumulation of cytoplasmic TFEB-GFP in the nucleus of both parental and Tam-resistant MCF7 cells, which indicates that neither the cytoplasmic retention nor the nuclear import mechanisms are altered as a consequence of the acquisition of Tam resistance. In addition, the nuclear export of TFEB, known to rely on the activity of the exportin XPO1/CRM1, normally operates in both normal and Tam-resistant cells, as demonstrated by the increased nuclear retention of the transcription factor determined by the XPO1 selective inhibitor LMB. Rather, our observations totally agree with the previous data reporting that Tam is capable of affecting the subcellular localization of TFEB in both breast cancer cells and in fibroblasts [13,50]. Our findings extend the previous observations to several additional breast cancer cell lines and demonstrate that such an atypical response of TFEB to Tam is independent of the intrinsic susceptibility or resistance of the cells to the drug. Consistently, the homogeneous pattern of TFEB relocation in the different breast cancer cell lines raises the possibility that this represents a sort of a common response of the transcription factor to Tam treatment of breast cancer cells of the luminal A subtype.

Another critical factor dictating the subcellular compartmentalization of TFEB is intracellular Ca^2+^, which, by regulating the activity of the protein phosphatase calcineurin, controls the subcellular positioning and transcriptional activity of TFEB [51,52]. Recently, the importance of even small, spatially restricted fluctuations in [Ca^2+^]_i_ in the activation of TFEB and of autophagy has been highlighted [19,20,27]. In particular, nutrient deprivation has been demonstrated to trigger the release of lysosomal Ca^2+^ through TRPML1, one of the most important lysosomal Ca^2+^ channels [53]. Also, another effective autophagy activator, rapamycin, has been demonstrated to directly bind TRPML1 and to induce the release of lysosomal Ca^2+^ [54], eventually increasing the nuclear relocation of TFEB and the autophagic flux. These findings demonstrate that the release of lysosomal Ca^2+^ represents a factor essential for TFEB activation and cellular adaptation to changes in the availability of nutrients. Along this line of evidence, any alteration or malfunctioning of the lysosomal compartment may perturb Ca^2+^ homeostasis and thereby impair the intracellular functions regulated by lysosomal Ca^2+^ [21,55]. As demonstrated by our previous report, Tam causes the permeabilization of the lysosomal membrane and the release of lysosomal contents, such as lysosomal cathepsins, to the cytoplasm [10]. Our present results first demonstrate that the mechanisms governing the nucleo-cytoplasmic shuttling of TFEB are normally operating in Tam-resistant cells, and that Tam triggers the release of lysosomal Ca^2+^ through the major lysosomal Ca^2+^ channels TRPML1 and TPC.

Indeed, the capability of Tam to alter intracellular Ca^2+^ homeostasis and signaling by acting through estrogen receptor-dependent or -independent mechanisms has already been evidenced by other research groups [25,26]. In the present study, we shed light on the fine mechanisms underlying a Tam-induced increase in [Ca^2+^]_i_. We first confirm the previous observations showing that Tam elicits a biphasic elevation in [Ca^2+^]_i_ that comprises a rapid Ca^2+^ peak decaying to a lower amplitude plateau and is initiated by GPER1 activation, as previously reported in MCF7 cells [25,26]. GPER1 is a G_s_-coupled membrane-associated estrogen receptor that elicits an array of intracellular signaling pathways, including an elevation in [Ca^2+^]_i_, in breast cancer cells [56]. Nevertheless, the mechanism whereby GPER1 activation leads to intracellular Ca^2+^ signaling has never been clearly elucidated. Intriguingly, a recent investigation demonstrated that GPER1 may also signal the InsP_3_-dependent increase in [Ca^2+^]_i_ by coupling to phospholipase C through G_q_ proteins [57]). Here, we find for the first time that Tam elicits ER Ca^2+^ release through InsP_3_Rs and lysosomal Ca^2+^ release through TPCs and TRPML1. In addition, the Ca^2+^ response to Tam is curtailed in the absence of extracellular Ca^2+^ and upon pharmacological blockade of SOCE, which represents the primary Ca^2+^ entry pathway activated by extracellular stimulation in MCF7 cells [30,31]. Overall, these findings support a mechanistic model according to which: (1) Tam stimulates GPER1 to recruit the signaling pathways that lead to the production of InsP_3_ (i.e., via G_q_ protein-driven phospholipase C recruitment), which gates InsP_3_Rs [58], nicotinic acid adenine dinucleotide phosphate (NAADP) (i.e., CD38 or DUOX2), which gates TPCs [22,59] and likely phosphatidylinositol-3,5-bisphosphate [PI(3,5)P2] (i.e., PIKfyve), which gates TRPML1 [22,60]; (2) InsP_3_Rs, TPCs and TRPML1 contribute to mediate the initial increase in [Ca^2+^]_i_ induced by Tam; (3) intraluminal Ca^2+^ efflux through InsP_3_Rs, in turn, causes a depletion of ER Ca^2+^ concentration, which leads to SOCE activation on the plasma membrane. Early studies showed that G_s_-coupled receptors, such as β1-adrenergic receptors, are coupled to NAADP production [61]. Therefore, Tam could stimulate GPER1 to recruit both G_s_- and G_q_-dependent signaling pathways, as shown in [57]. By contrast, the signal transduction pathway coupling GPER1 to TRPML1 activation remains unclear, since the mechanisms by which G protein-coupled receptors are coupled to PIKfyve are yet to be elucidated [60].

These findings, therefore, extend the early observation that Tam-induced GPER1 activation leads to intracellular Ca^2+^ signaling in MCF7 cells by only activating InsP_3_Rs [34]. However, only the Tam-induced lysosomal Ca^2+^ release through TRPML1 and TPCs drives the calcineurin-mediated dephosphorylation and the ensuing nuclear relocation of TFEB, eventually culminating in the upregulation of TFEB itself and of its target genes [48], and in the enhancement of the autophagic flux. Recent findings showed that SOCE is also able to drive the nuclear translocation of TFEB [62,63]. Since SOCE is activated downstream of InsP_3_Rs-mediated ER Ca^2+^ depletion, this observation suggests that InsP_3_Rs can be indirectly coupled to TFEB activation in other cell types. However, the pharmacological blockade of InsP_3_Rs and SOCE with 2-APB and BTP-2, respectively, neither prevented Tam-induced TFEB translocation into the nucleus nor affected MCF7-TamR cell viability (Figure 8). Therefore, we are reasonably confident that lysosomal TPCs and TRPML1 are the primary sources of Ca^2+^ responsible for the nuclear translocation of TFEB in luminal A breast cancer cells.

Since the nuclear localization of TFEB is increased in MCF7-TamR cells, we reasoned that this was due to an elevation in resting [Ca^2+^]_i_ consequent to Tam-dependent GPER1 activation. Accordingly, we found that basal [Ca^2+^]_i_ was significantly higher in MCF7-TamR as compared to MCF7 cells, but that resting Ca^2+^ levels were restored by inhibiting GPER1 with G15 or upon Tam washout from the perfusate. In agreement with these findings, Tam fails to elicit a robust Ca^2+^ signal in MCF7-TamR cells after a brief removal of the agonist, which is consistent with the partial desensitization of GPER1 after prolonged stimulation [64]. Intriguingly, the pharmacological blockade of either InsP_3_Rs with 2-APB or of SOCE with BTP-2 did not affect the nuclear localization of TFEB in MCF7-TamR cells. Only the inhibition of lysosomal Ca^2+^ release via TPCs (with NED-19) or via TRPML1 (with ML-SI3) suppressed the Tam-induced nuclear relocation of TFEB. These findings strongly support the view that the Tam-induced release of lysosomal Ca^2+^ is the main molecular mechanism accounting for the greater nuclear relocation of TFEB detected in MCF7-TamR cells stably growing in the presence of the drug, as well as in the parental breast cancer cell lines transiently treated with the drug. Furthermore, our results strongly indicate that the mechanisms governing the subcellular localization and biological activity of TFEB in response to Tam treatment are shared by all the breast cancer cell lines tested, irrespectively of their susceptibility or resistance to the drug.

Interestingly, the Tam-induced elevation in the autophagic flux deriving from the mechanisms delineated above is intimately connected with the enhanced resistance of MCF7-TamR cells to Tam. In fact, stimulation of autophagy is known to favor the survival of cancer cells to anticancer treatments [10], but how this takes place at the molecular level is still far from fully elucidated. On this, we and others have previously demonstrated that Tam, in addition to its main action on the estrogen receptor, may oxidatively damage the lysosomal membrane [10], which may progress until its severe permeabilization, in this way triggering a lysosome-mediated death pathway (for a review, see [65]). Our previous results have demonstrated that the increased autophagic flux in MCF7-TamR cells enhances the lysosomal delivery of cytoplasmic factors, essentially iron-binding proteins [10], effective in restraining drug-induced intralysosomal oxidative stress. The latter, by preventing lysosomal permeabilization and the ensuing activation of lysosomal death pathways, favors cells’ survival and resistance to anticancer treatment(s).

In the context of the essential role played by autophagy in driving anticancer drug resistance, pharmacological inhibition of the autophagic flux has been proposed and tested in clinical settings as a suitable strategy to prevent the development of autophagy-mediated drug resistance or to restore the susceptibility of cancer cells to therapeutic agents [66,67,68]. The evidence that lysosomal TRPML1 and TPCs drive the nuclear translocation of TFEB in Tam-resistant cells hints at an alternative strategy to circumvent endocrine resistance in breast cancer of the luminal A subtype. Accordingly, TPCs are under intense scrutiny as potential druggable targets in cancer patients [22,69] as well as in other disorders, including cardiovascular diseases [70] and COVID-19 [71]. Preclinical studies have shown that NED-19 reduced the formation of lung metastases in 4T1 breast cancer [72] and B16 melanoma [73] xenograft models in vivo. Of note, the i.p. administration of NED-19 did not exert overt off-side targets over a 4-week period [72]. Additional preclinical studies are obviously required before translating NED-19 into clinical applications. However, TPCs are also sensitive to many FDA-approved drugs that have been repurposed as TPCs inhibitors, including verapamil, diltiazem, fluphenazine and pimozide [71,74]. Therefore, it has been proposed that these drugs could also be tested against disorders associated with TPCs-mediated lysosomal Ca^2+^ release, including cancer [22,69]. Less information is available regarding the potential off-target effects of TRPML1 inhibition and the available FDA-approved TRPML1 blockers. In addition to ML-SI3, TRPML1 can be inhibited by ML-SI1 [38] and the steroidal compound estradiol methyl ether [75]. The pharmacological blockade of TRPML1 has been proposed as alternative approach to treat many diseases associated with aberrant TRPML1-mediated Ca^2+^ release, including cancer [76], lysosomal storage disorders [77,78] and inflammatory disorders [79]. It has recently been proposed that improving the specificity, potency and efficacy of ML-SI13 and ML-SI1 by using medicinal chemistry will be helpful for the therapeutic translation of TRPML1 in cancer.

This study presents two potential limitations. The first is that this investigation focuses on the identification of the mechanisms by which Tam activates autophagy specifically in breast cancer cell lines of the luminal A subtype. Due to the specificity of the experimental model, there is the possibility that the mechanisms of Tam-induced activation of autophagy identified in these cells do not overlap with those triggered by different anticancer drugs commonly used for the therapy of other subtypes of breast cancer. The second limitation may be represented by the type of anticancer drug used. Although the Tam-induced mechanism leading to autophagy activation is relevant for luminal A breast cancer cells, it should be verified with other drugs which are known to elevate the autophagic flux in breast cancer cells of different molecular subtypes. Although extremely relevant both from the theoretical point of view and for the potential clinical applications, these aspects are beyond the scope of the present investigation and will be addressed in further studies.

The results gathered in the present investigation demonstrate that Tam rapidly and reversibly triggers the nuclear relocation of TFEB in breast cancer cells of the luminal A subtype by triggering lysosomal Ca^2+^ mobilization through TPCs and TRPML1, which therefore emerge as potential new molecular targets to prevent Tam resistance in breast cancer.

These observations provide the first mechanistic explanation of how Tam orchestrates the activation of protective autophagy in luminal A breast cancer cells, which ultimately may lead to endocrine resistance. Further investigations will help to clarify whether the alteration of intracellular dynamics of TFEB represents a specific consequence of Tam treatment on luminal A breast cancer cells or rather a prototypical strategy of tumor cells to resist to anticancer treatments.

## 4. Materials and Methods

### 4.1. Chemicals

DMEM (D6429), fetal bovine serum (FBS, F7524), penicillin/streptomycin solution (P0781), geneticin (G1421), LMB (L2913), MTT (M2128) were from Merck (Darmstadt, Germany). Tam (sc-208414) was from Santa Cruz Biotechnology (Heidelberg, Germany).

Fura-2 acetoxymethyl ester (Fura2, F1221) was from Life Technologies (Thermo Fisher Scientific, Waltham, MA, USA). 4-methyl-4′-[3,5-*bis*(trifluoromethyl)-1H-pyrazol-1-*yl*]-1,2,3-thiadiazole-5-carboxanilide (BTP-2), (1R,3S)-1-[3-[[4-(2-fluorophenyl)piperazin-1-yl]methyl]-4-methoxyphenyl]-2,3,4,9-tetrahydro-1H-pyrido [3,4-b]indole-3-carboxylic acid (NED-19), (±)-1-[(3a*R**,4*S**,9b*S**)-4-(6-Bromo-1,3-benzodioxol-5-yl)-3a,4,5,9b-tetrahydro-3*H*-cyclopenta[*c*]quinolin-8-yl]-ethenone (G1), and (3a*S**,4*R**,9b*R**)-4-(6-Bromo-1,3-benzodioxol-5-yl)-3a,4,5,9b-3*H*-cyclopenta[*c*]quinoline (G15) were from Tocris Bioscience (Bristol, UK). 2-aminoethyl diphenylborinate (2-APB) and 1-(2,3-dichlorobenzoyl)-5-methoxy-2-methyl-(3-(morpholin-4-yl)ethyl)-1H-indole hydrochloride (ML-SI3) were from Sigma-Aldrich (Saint Louis, MO, USA).

### 4.2. Cell Cultures

MCF7, MCF7-TamR [2], MDA-MB-415, T47D and ZR-75-1 cells were grown in DMEM supplemented with 10% FBS, 100 U/mL penicillin and 100 μg/mL streptomycin at 37 °C in a humidified atmosphere containing 5% CO_2_. For ZR-75-1 and ZR-75-1-TamR cells, 1% nonessential amino acids were added to the culture medium. Tam-resistant MDA-MB-415, T47D and ZR-75-1 cells were generated by growing the parental cells in the presence of increasing concentrations of the drug as previously described [10] and designated by adding the suffix ‘-TamR’ to the name of their parental counterpart. Unless differently indicated, all the Tam-resistant cell lines were routinely maintained in the presence of 5 µM Tam in the growth medium.

### 4.3. Cell Growth and Viability Assay

For the determination of the growth kinetics, the cells were seeded in 24-well plates (Jet Biofil, Alicante, Spain) at 2 × 10^4^ cells/cm^2^ and grown for 4 or 6 days, when they were detached by trypsinization and counted. Where required, 5 μM Tam was added at seeding time and replaced every third day.

Cell viability was determined with the MTT test. MCF7 and MCF7-TamR were seeded at 3 or 3.5 × 10^4^/cm^2^, respectively, in 100 µL of growth medium; after treatments, 20 µL of an MTT solution (5 mg/mL) was added to each well for 2 h. The formazan precipitates were dissolved in 100 µL of dimethyl sulfoxide by shaking the plates over an orbital shaker for 30 min at room temperature before measurement of the absorbance at 595 nm with the iMark Microplate Reader (Bio-Rad Laboratories, Hercules, CA, USA).

### 4.4. Immunofluorescence

To investigate the subcellular localization of endogenous TFEB, MCF7 and MCF7-TamR cells were seeded on a µ-Slide 8 well (80826, Ibidi, Gräfelfing, Germany) at 4 and 4.5 × 10^4^/cm^2^, respectively, and allowed to grow for 48 h. Where required, cells were treated with 5 µM Tam for 24 h. After fixation with cold methanol-acetone (1:1 *v*/*v*) for 10 min at −20 °C, the cells were rehydrated with phosphate-buffered saline (PBS) containing 0.05% Triton and 0.05% NaN_3_ and subsequently incubated overnight at 4 °C with an antibody anti-TFEB (1:200, sc-166736, Santa Cruz Biotechnology). The cells were then washed three times with PBS, incubated with 2 µg/mL of an Alexa Fluor-conjugated goat anti-mouse secondary antibody (A-21424, Thermo Fisher Scientific) for 2 h at room temperature in the dark and washed twice with distilled sterile water. Confocal microscopy analysis, image acquisition and processing were performed as described below (see Section 4.13).

### 4.5. Transient and Stable Transfection of TFEB-GFP

For transient transfection, the cells were seeded at 3 × 10^4^/cm^2^ (MCF7 and MCF7-TamR) or 4 × 10^4^/cm^2^ (MDA-MB-415, T47D and ZR-75-1) in 24-well tissue culture plates (Jet Biofil) and transfected 24 h later with a plasmid encoding human TFEB-GFP [a gift from Shawn Ferguson (Addgene plasmid #38119)] using the K2 Transfection Reagent (T060-0.75, Biontex, München, Germany). The next day, the transfection medium was substituted with fresh medium for additional 72 h; during this time only, Tam was omitted from the growth medium of all TamR cells to avoid excessive toxicity. Subsequently, the cells were treated as indicated in the results, fixed with 4% paraformaldehyde and stored at 4 °C in the dark until confocal microscopy analysis.

For stable transfections, MCF7 and MCF7-TamR cells were seeded as for transient transfection on 60 mm Petri dishes and transfected with TFEB-GFP in the absence of Tam as described above. The drug was added again to MCF7-TamR cells 72 h after the removal of the transfection medium. Forty-eight hours later, 400 μg/mL geneticin was added to the growth medium for 3 weeks, with antibiotic replacement every 3 days. After this time, cells stably growing in the presence of geneticin were used for the experiments.

### 4.6. Determination of Nuclear Localization of TFEB

Confocal microscopy images of both TFEB immunofluorescence and of cells transfected with TFEB-GFP were used to quantify the percentage of cells showing nuclear translocation of the transcription factor. To make the identification of such cells objective and operator-independent, the Intensity Profile plugin of the Icy software, version 2.4.3.0 [80], was used. To measure the basal nuclear fluorescence intensity, a linear region of interest (ROI) was drawn across the major diameter of at least fifteen control nuclei (MCF7 cells in basal growth conditions) devoid of any evident nuclear localization of TFEB. The fluorescence along any single ROI was recorded and used to calculate the mean fluorescence intensity. Only the cells with nuclei whose fluorescence was greater than the mean + 2 SD of the basal fluorescence of controls were considered to have nuclear TFEB.

### 4.7. Analysis of the Effects of Starvation or Tam on the Subcellular Localization of TFEB and Washout Experiments in TFEB-GFP-Expressing Breast Cancer Cell Lines

MCF7 cells were transfected with TFEB-GFP as described above. To monitor the effects of the drug on the subcellular relocation of TFEB, 72 h after the transfection, 5 µM Tam was added in the growth medium for additional 24 h. For Tam washout, the Tam-containing medium was replaced with the same volume of fresh medium lacking the drug for further 24 h, either in the absence or presence of 10 ng/mL LMB [50,81], before fixation and confocal analysis.

MCF7-TamR cells were plated at 3.5 × 10^4^ cells/cm^2^ in medium without Tam, allowed to adhere overnight and transfected with TFEB-GFP as described above. Seventy-two hours after transfection, Tam was added to the cells for 24 h; MCF7-TamR cells in these experimental conditions were taken as controls for the study of the Tam-induced subcellular localization of TFEB. For the washout experiments, Tam was subsequently removed for 24 h by a medium change either in the absence or presence of 10 ng/mL LMB, before fixation and confocal analysis. To investigate the effects of starvation on the shuttling of TFEB between the cytoplasm and the nucleus, MCF7-TamR cells were transfected with TFEB-GFP in the absence of Tam as above; the drug was then omitted for the whole duration of both starvation with HBSS-glucose and refeeding to avoid interferences due to the capability of Tam to impact the subcellular localization of the transcription factor. Where required, LMB was added 30 min before refeeding and kept for all the subsequent refeeding time.

### 4.8. Solutions for Ca^2+^ Recordings

Physiological salt solution (PSS) had the following composition (in mM): 150 NaCl, 6 KCl, 1.5 CaCl_2_, 1 MgCl_2_, 10 Glucose, 10 HEPES. In Ca^2+^-free solution (0Ca^2+^), Ca^2+^ was substituted with 2 mM NaCl and 0.5 mM EGTA was added. Solutions were adjusted to pH 7.4 with NaOH. The osmolality of PSS as measured with an osmometer (Wescor 5500, Logan, UT, USA) was 338 mmol/kg.

### 4.9. Intracellular Ca^2+^ Imaging

Tam- and G1-induced changes in [Ca^2+^]_i_ were monitored in MCF7 and MCF7-TamR cells loaded with the selective Ca^2+^-fluorophore Fura-2 (4 µM; 1 mM stock in dimethyl sulfoxide) in PSS for 30 min at 37 °C and 5% CO_2_, as shown in [32]. The cells were plated on round glass coverslips (8 mm diameter) coated with collagen (5 mg/mL; Sigma Aldrich), bathed with PSS, loaded with 4 µM Fura-2 and then maintained in the presence of the Ca^2+^ indicator for 30 min at 37 °C and 5% CO_2_. Subsequently, the cells were extensively washed with fresh PSS and the coverslip was gently attached to the bottom of a Petri dish with silicon grass (Saratoga, Trezzano sul Naviglio, Milan, Italy). The Petri dish was then moved on the stage of an upright epifluorescence Axiolab microscope (Carl Zeiss, Oberkochen, Germany) and the cells were observed with a Zeiss ×40 Achroplan objective (water-immersion, 0.9 numerical aperture, 2.0 mm working distance). Every 3 sec, Fura-2 was alternately (0.5 Hz) excited at 340 and 380 nm, and the emitted fluorescence was recorded at 510 nm. A filter wheel (Lambda 10, Sutter Instrument, Novato, CA, USA) was used to accommodate the excitation filters. From 10 to 40 rectangular ROIs were drawn around the cells that were clearly identifiable in the visual field. At each excitation wavelength, images of the visual field and the fluorescence within each ROI were acquired by an Extended-ISIS Camera (Photonic Science, Millham, UK). A custom software that worked in the LINUX environment was employed to control both the Extended-ISIS Camera and the filter wheel. The LINUX-based software (version 1.2.0) was also used to measure the ratio of the mean fluorescence emitted at 510 nm when the cells within each ROI were excited alternatively at 340 and 380 nm (F340/F380). The amplitude of cytosolic Ca^2+^ signals evoked by each agonist (Tam and G1) was measured as the difference between the F340/F380 ratio at the peak of the Ca^2+^ signal and the mean F340/F380 ratio of 1 min baseline recorded before addition of the agonist. All recordings were carried out at room temperature (22 °C).

### 4.10. Analysis of Differential Gene Expression in MCF7 and MCF7-TamR Cells

Total RNA was extracted from MCF7 and MCF7-TamR cells with the ReliaPrep™ RNA Miniprep System (Z6010, Promega, Madison, WI, USA) according to the supplier directions, and used for strand-specific library preparation and RNA-seq reactions at Novogene (Cambridge, UK). RNA sequencing was performed with the Illumina platform and was followed by quality control analysis. The resulting reads were aligned to the GRCh38 reference human genome using STAR software (version 2.5) to obtain the read counts for the downstream analyses. To analyze the raw data, R analytical software (RStudio Team, RStudio, Boston, MA, USA) was used under the graphical user interface designated as Rstudio (version 4.0.5, 2021). Differential gene expression analysis was performed with DESeq2 package (version 1.40.2) in Bioconductor (version 3.17).

### 4.11. Expression of TFEB Target Genes

Total RNA was extracted with the TriReagent (T9424, Merck); 1 µg of total RNA from each sample was reverse transcribed with the FireScript RT cDNA synthesis Kit (06-12-00200, Solis BioDyne, Tartu, Estonia). A volume of cDNA corresponding to 50 ng of total RNA/sample was amplified in a CFX Connect (Bio-Rad Laboratories) with the HOT FIREPol Evagreen qPCR Supermix (08-36-00001, Solis BioDyne). The PCR primers used for the amplifications were designed with Primer 3 (Table 1). The relative mRNA content was calculated using the 2^−∆∆CT^ method.

### 4.12. Western Blotting

For Western blotting analysis, the cells were detached by trypsinization and collected by centrifugation at 600× *g* for 10 min. The samples were homogenized by sonication in ice-cold RIPA buffer for 10 sec; 50 µg of total lysates was separated on a 10% polyacrylamide gel and blotted to a nitrocellulose membrane. Probing of the membranes with anti-TFEB (A303-673A, Bethyl Laboratories, Montgomery, TX, USA) and anti-β-actin (A5441, Merck) antibodies was performed as described [10]. Bands were detected with the ChemiDoc XRS+ Imaging System (Bio-Rad Laboratories). The optical density of the TFEB band was normalized against that of β-actin.

### 4.13. Confocal Microscopy

Parental or Tam-resistant cells were seeded on a µ-Slide 8 well (80826, Ibidi) at 3 or 3.5 × 10^4^/cm^2^, respectively, and transfected with the plasmid encoding TFEB-GFP as described above. After the treatments, the cells were washed twice with PBS, fixed with 4% paraformaldehyde and washed twice with sterile H_2_O. Samples were subsequently observed with a LSM800 confocal microscope (Carl Zeiss). Images were processed with ImageJ. The fraction of cells with nuclear TFEB (see ‘Determination of nuclear localization of TFEB’) was obtained by examining 90–250 cells for each experimental condition from images taken from different microscopic fields.

### 4.14. Downregulation of TFEB or TFE3

Silencing of TFEB or TFE3 was achieved by transfecting the cells with the predesigned MISSION^®^ esiRNAs (EHU059261 or EHU157921, respectively; Eupheria Biotech, Merck) with Lipofectamine RNAiMAX (13778100, Thermo Fisher Scientific) for 72 h; effectiveness of silencing was verified by relative RT-qPCR. The final concentration of each esiRNA was 40 nM; when both TFEB and TFE3 were silenced in the same sample, the concentration of each esiRNA was halved.

### 4.15. Statistical Analysis

Data represent the mean ± SD of three independent experiments performed in triplicate. Unless differently indicated, differences between groups were assessed with either One-Way ANOVA followed by Dunnett’s or Tukey’s post hoc as appropriate, or Student’s *t*-test using the Instat package (Version 3.10, GraphPad Software, Boston, MA, USA). A *p* < 0.05 was considered statistically significant.

For Ca^2+^ measurements, pooled data are presented as mean ± SE. The number of cells analyzed for each condition is indicated in the corresponding histograms. Differences between two groups were evaluated by using Student’s *t*-test for unpaired observations, whereas differences between multiple groups were evaluated using One-Way ANOVA analysis followed by post hoc Dunnett’s or Bonferroni tests as appropriate. *p* < 0.05 indicated statistical significance.

## Figures and Tables

**Figure 1 ijms-25-00458-f001:**
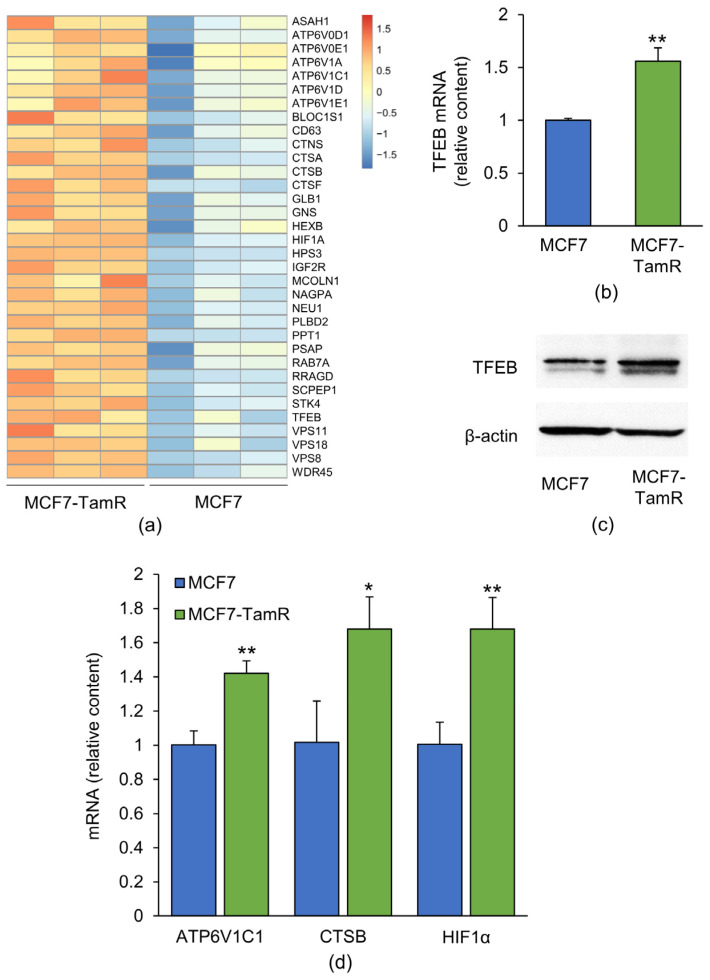
Expression of TFEB and of its target genes in MCF7 and MCF7-TamR cells. (**a**) Heat map showing the mRNA amount of TFEB and of its target genes upregulated in MCF7-TamR cells, as obtained from the RNA-seq data. Each column represents a sample. Relative amount of TFEB mRNA (**b**) and protein (**c**) in MCF7 and MCF7-TamR cells. (**d**) Real-Time RT-qPCR validation of the RNA-seq data for selected TFEB targets involved in autophagy. The picture in panel (**c**) is representative of three independent experiments. Data represent the mean ± SD of three independent experiments. Statistical significance was assessed with Student’s *t*-test; *: *p* < 0.05; **: *p* < 0.01.

**Figure 2 ijms-25-00458-f002:**
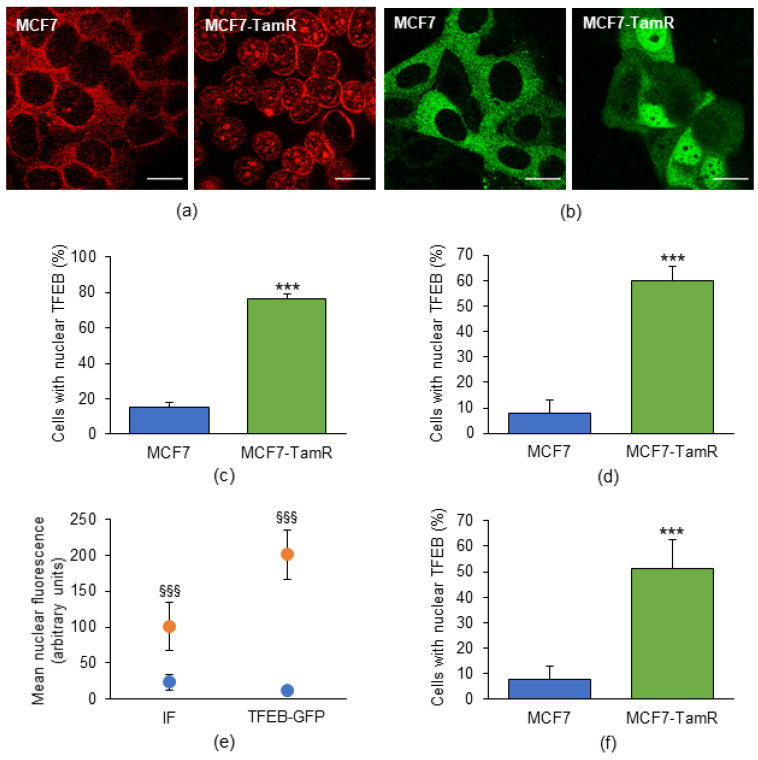
Subcellular localization of TFEB in MCF7 and MCF7-TamR cells as detected using different techniques. The panels show representative confocal images of both MCF7 and MCF7-TamR cells in basal growth conditions, either stained with immunofluorescence (**a**) or following transient transfection with TFEB-GFP (**b**). (**c**,**d**) Fraction of cells displaying nuclear TFEB, as determined by immunofluorescence or transient transfection of TFEB-GFP, respectively. (**e**) Mean fluorescence intensity of nuclei without or with nuclear TFEB (blue or orange dots, respectively), from samples prepared with the indicated techniques. (**f**) Fraction of cells with nuclear TFEB as calculated in cells stably transfected with TFEB-GFP. IF, immunofluorescence; TFEB-GFP, cells transiently transfected with the TFEB-GFP chimera. Data represent the mean ± SD of three independent experiments. Statistical significance was assessed with Student’s *t*-test; ***: *p* < 0.001; §§§: *p* < 0.001 vs. the corresponding control nuclei. Scale bar: 20 µm.

**Figure 3 ijms-25-00458-f003:**
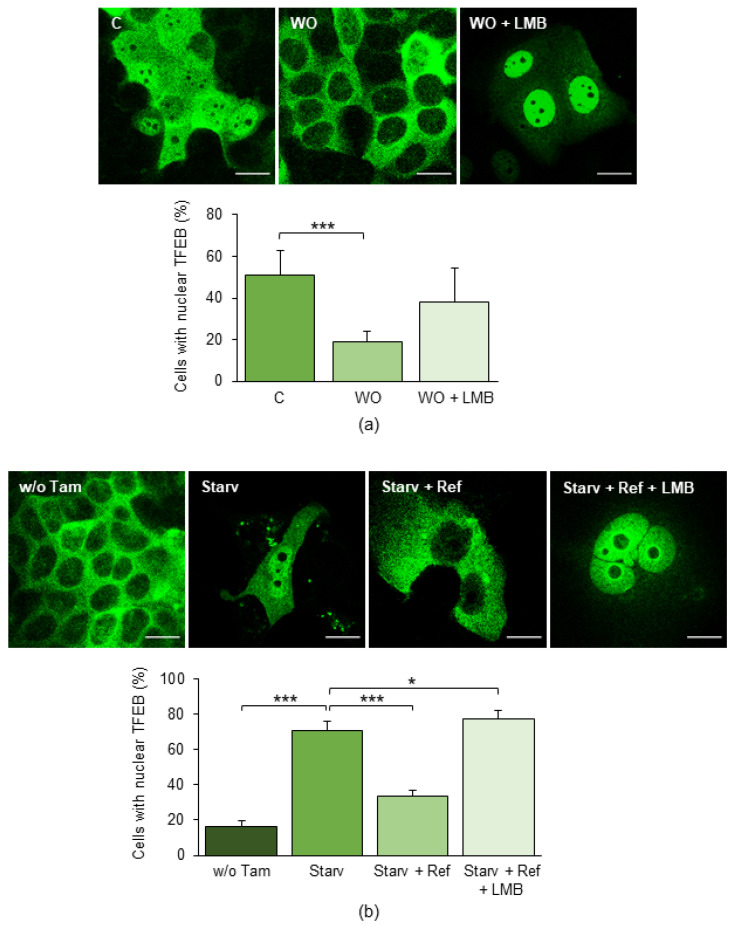
Effects of Tam washout and starvation on the subcellular localization of TFEB-GFP in MCF7-TamR cells. (**a**) Cells treated with 5 µM Tam as indicated in Section 4 or after Tam removal (washout) either in the absence or presence of 10 ng/mL of leptomycin B (LMB). (**b**) Cells grown in the absence of Tam starved for 4 h in HBSS-glucose and subsequently refed for 2 h with complete growth medium in the absence or presence of 10 ng/mL of LMB. The bar charts below each set of images represent the percentage of cells with nuclear TFEB. Starv: starvation; Ref: refeeding; LMB: 10 ng/mL leptomycin B; Tam: 5 µM tamoxifen; WO: washout; w/o Tam: MCF7-TamR cells grown in the absence of Tam to avoid Tam-induced TFEB relocation events. Data represent the mean ± SD of three independent experiments. Statistical significance was assessed using ANOVA followed by Dunnett’s post hoc test; *: *p* < 0.05; ***: *p* < 0.001. Scale bar: 20 µm.

**Figure 4 ijms-25-00458-f004:**
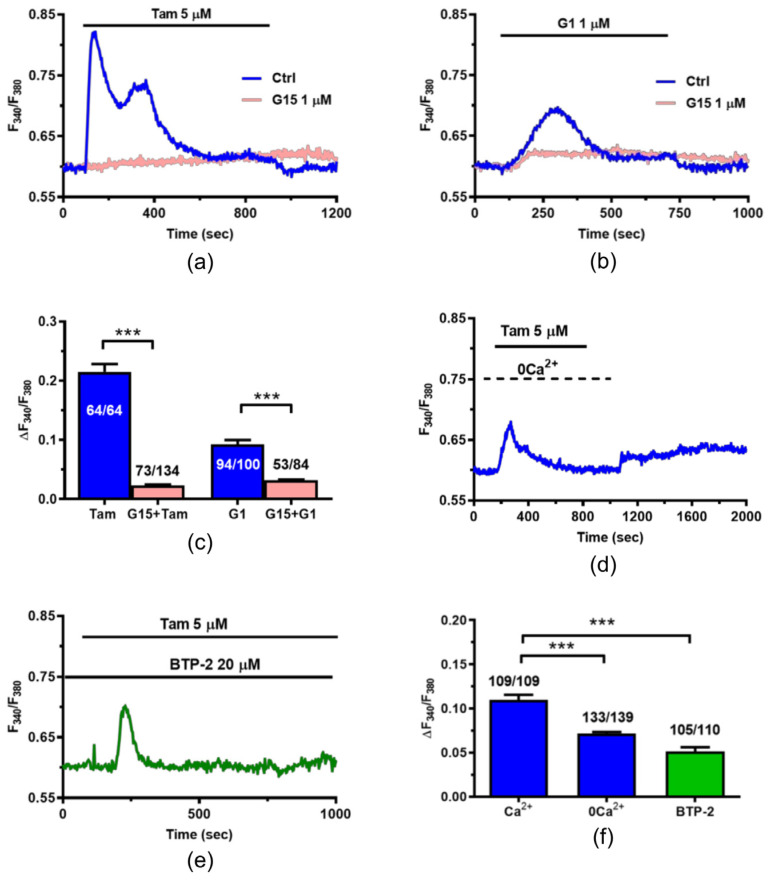
Tam induces an increase in [Ca^2+^]_i_ in MCF7 cells. (**a**) Tam (5 µM) elicits a biphasic elevation in [Ca^2+^]_i_ in MCF7 cells loaded with the Ca^2+^-sensitive fluorophore, Fura-2. The Ca^2+^ response to Tam (Ctrl) is inhibited by G15 (1 µM), a selective GPER1 blocker. (**b**) The specific GPER1 agonist, G1 (1 µM), elicits a biphasic increase in [Ca^2+^]_i_ in MCF7 that is blocked by G15 (1 µM). (**c**) Mean ± SE of the amplitude of the initial Ca^2+^ peak elicited by either Tam or G1 in the absence and presence of G15. Student’s *t*-test: ***: *p* < 0.001. (**d**) In the absence of extracellular Ca^2+^ (0Ca^2+^), Tam (5 µM) elicits a transient increase in [Ca^2+^]_i_. The subsequent restoration of extracellular Ca^2+^ concentrations (1.5 mM) elicits a second increase in [Ca^2+^]_i_ that is indicative of SOCE activation. (**e**) In the presence of BTP-2 (20 µM), a selective blocker of SOCE, Tam (5 µM) elicits a transient increase in [Ca^2+^]_i_. (**f**) Mean ± SE of the amplitude of the initial Ca^2+^ peak elicited by Tam in the presence (Ca^2+^) or absence of extracellular Ca^2+^ (0Ca^2+^) and in the presence of BTP-2. One-Way ANOVA followed by the post hoc Dunnett’s test: ***: *p* < 0.001.

**Figure 5 ijms-25-00458-f005:**
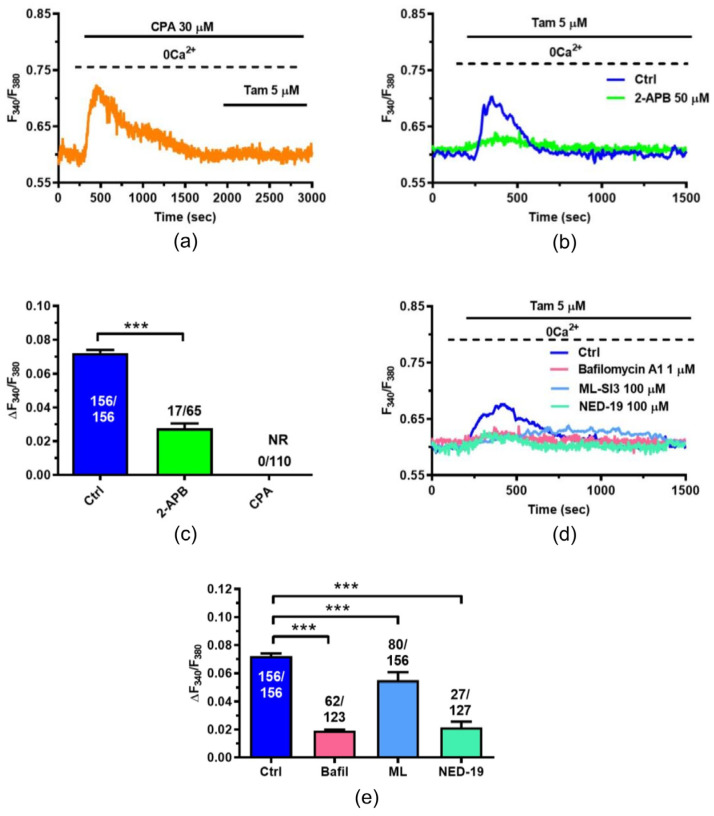
InsP3Rs and TPCs support Tam-induced intracellular Ca^2+^ release in MCF7 cells. (**a**) Tam (5 µM) fails to elicit a detectable elevation in [Ca^2+^]_i_ upon depletion of the ER Ca^2+^ store with CPA (30 µM), a selective inhibitor of SERCA activity. As expected, CPA induces a transient Ca^2+^ signal which is due to passive ER Ca^2+^ efflux through yet-to-be-determined ER leakage channels. (**b**) Tam (5 µM) elicits a transient increase in [Ca^2+^]_i_ in the absence (Ctrl) but not in the presence of 2-APB (50 µM), a selective InsP3R blocker. (**c**) Mean ± SE of the amplitude of the Ca^2+^ peak elicited by Tam in the absence (Ctrl) and presence of 2-APB and CPA. One-Way ANOVA followed by the post hoc Dunnett’s test: ***: *p* < 0.001. (**d**) Tam (5 µM) elicits robust intracellular Ca^2+^ release in the absence (Ctrl) but not in the presence of the following drugs: Bafilomycin A1 (1 µM), which depletes the lysosomal Ca^2+^ store by inhibiting the activity of v-ATPase; ML-SI3 (100 µM), which blocks TRPML1; and NED-19 (100 µM), which blocks TPCs. (**e**) Mean ± SE of the amplitude of the Ca^2+^ peak elicited by Tam in the absence (Ctrl) and presence of Bafilomycin A1 (Bafil), ML-SI3 (ML) and NED-19. One-Way ANOVA followed by the post hoc Dunnett’s test: ***: *p* < 0.001.

**Figure 6 ijms-25-00458-f006:**
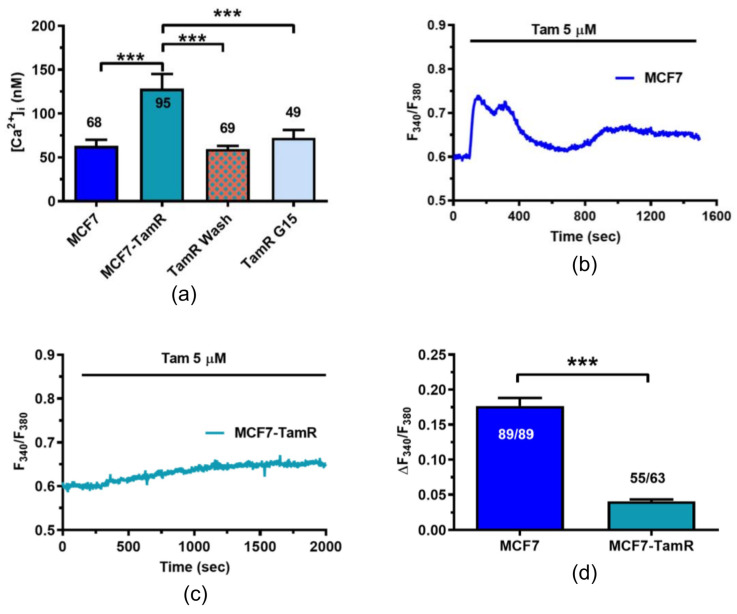
Resting [Ca^2+^]_i_ is higher in MCF7-TamR due to persistent GPER1 activation. (**a**) Mean ± SE of resting [Ca^2+^]_i_ measured in MCF7 cells, MCF7-TamR cells, MCF7-TamR cells at 48 h after Tam washout and MCF7-TamR cells bathed with G15 (1 µM) for 48 h. One-Way ANOVA followed by the post hoc Dunnett’s test: ***: *p* < 0.001. (**b**) Tam (5 µM) elicits a robust increase in [Ca^2+^]_i_ in MCF7 cells. (**c**) Tam (5 µM) elicits a weak increase in [Ca^2+^]_i_ in MCF7-TamR cells. (**d**) Mean ± SE of the amplitude of the Ca^2+^ increase elicited by Tam in MCF7 cells and MCF7-TamR cells. Student’s *t*-test: ***: *p* < 0.001.

**Figure 7 ijms-25-00458-f007:**
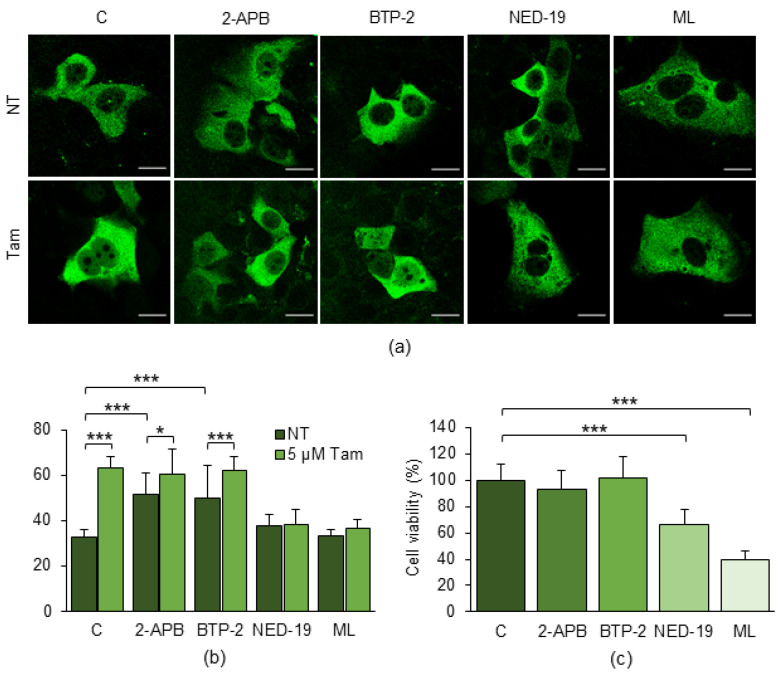
Effects of inhibitors of Ca^2+^ channels on Tam-induced nuclear relocation of TFEB. (**a**) Representative confocal images of MCF7-TamR cells treated or not with 5 µM Tam for 24 h in the absence or presence of the indicated inhibitors of the Ca^2+^-permeable channels. (**b**) Percentage of MCF7-TamR cells displaying nuclear TFEB, either in the absence or presence of 5 µM Tam. (**c**) Viability of MCF7-TamR cells treated for 5 days with the inhibitors of Ca^2+^ channels (50 µM 2-APB, 20 µM BTP-2, 100 µM NED-19, 5 µM ML-SI). NT: Tam-resistant cells cultured in the absence of Tam; ML: ML-SI3. Data represent the mean ± SD of three independent experiments. Statistical significance was assessed using ANOVA followed by Tukey’s (**b**) or Dunnett’s (**c**) post hoc tests; *: *p* < 0.05; ***: *p* < 0.001. Scale bar: 20 µm.

**Figure 8 ijms-25-00458-f008:**
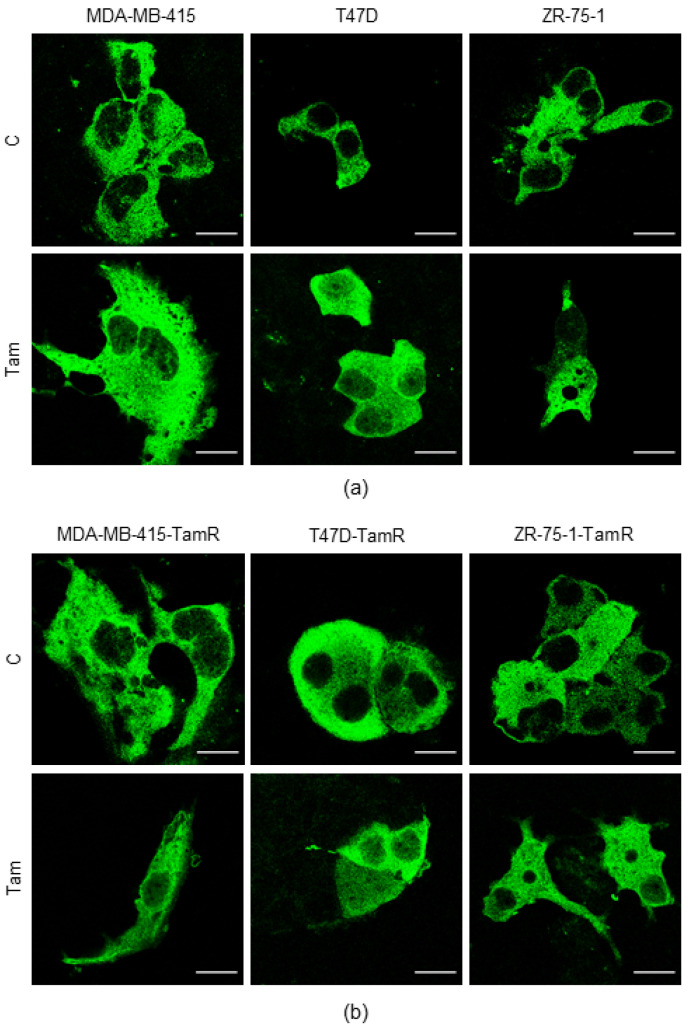
Effect of Tam on the subcellular localization of TFEB-GFP in parental and Tam-resistant cells. Representative images of subcellular localization of TFEB in parental (**a**) or Tam-resistant (**b**) MDA-MB-415, T47D and ZR-75-1 cells transiently transfected with a plasmid encoding TFEB-GFP, in normal growth condition (upper panels) or in the presence of 5 µM Tam for 24 h (lower panels). Tam: 5 µM Tam. Scale bar: 20 µm.

**Table 1 ijms-25-00458-t001:** Primers used for RT-qPCR.

Target	Forward	Reverse
Hs_ATP6V1C1	CCCGAGGAGTCTGCTGGTTC	AAGTGTGCCTCCCCTTCGAC
Hs_CTSB	TGGAAGCCATCTCTGACCGG	TACAGCCGTCCCCACACATG
Hs_HIF1α	CAGCTATTTGCGTGTGAGGA	CCTCATGGTCACATGGATGA
Hs_TFE3	TGCTGTTGGAGGAGCGCA	CTTGAGCGAAGGGGTAAGGG
Hs_TFEB	GTCCGAGACCTATGGGAACA	TCGTCCAGACGCATAATGTTG
Hs_GAPDH	CGGGAAACTGTGGCGTGATG	ATGCCAGTGAGCTTCCCGTT

## Data Availability

The datasets generated in the present study are not yet deposited in any public repository and are available from the corresponding author on reasonable request.

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
