# Peer review of "Tamoxifen Activates Transcription Factor EB and Triggers Protective Autophagy in Breast Cancer Cells by Inducing Lysosomal Calcium Release: A Gateway to the Onset of Endocrine Resistance"

_ijms, 2023, doi:10.3390/ijms25010458_

Round 1
Reviewer 1 Report
Comments and Suggestions for Authors
Trying to understand Tamoxifen resistance in luminal A breast cancer cells, the authors explored the underlying molecular mechanisms using MCF7 Tamoxifen – induced resistance cell line.
The authors found the major contribution of a transcription factor (TFEB that promotes expression of genes involved in lysosome biogenesis and autophagy) to Tam-induced resistance. The authors generated supporting data to demonstrate that Tam increased intracellular Ca2+ levels that drive the cytosol-to-nuclear translocation of TFEB. Increased intracellular Ca2+ levels are due to Ca2+ release from lysosomal Ca2+-releasing channel, two pore channel, and Ins3R at endoplasmic reticulum. The role of specific TFEB, no other related transcription factors, in Tam resistance is confirmed by silencing TFEB. The results are reproduced in additional three luminal A breast cancer cell lines. The experiments are well designed to support the conclusions.
Minor,
Line 186: “am”? Tam?
Comments on the Quality of English LanguageNA
Reviewer 2 Report
Comments and Suggestions for Authors
The paper entitled "Tamoxifen activates TFEB and triggers protective autophagy in breast cancer cells by inducing lysosomal calcium release: a gateway to the onset of endocrine resistance" by Cecilia Boretto, Chiara Actis, Pawan Faris, Francesca Cordero, Marco Beccuti, Giulio Ferrero, Giuliana Muzio, Francesco Moccia and Riccardo Autelli presents the investigation about how TFEB is activated and contributes to Tam resistance in luminal A breast cancer cells. Tam promoted the release of lysosomal Ca2+ through the major transient receptor potential cation channel mucolipin subfamily member 1 (TRPML1) and two-pore channels (TPCs), which caused the nuclear translocation and activation of TFEB. Consistently, inhibiting lysosomal calcium release restored the susceptibility of MCF7-TamR cells to Tam. Their findings demonstrate that Tam drives the nuclear relocation and transcriptional activation of TFEB by triggering the release of Ca2+ from the acidic compartment, and suggest that lysosomal Ca2+ channels may represent new druggable targets to counteract the onset of autophagy-mediated endocrine resistance in luminal A breast cancer cells.
The paper is well written and suported by experimental methods. Overall I apreciatte the quality of the study. regardless, there a re a few minor points I would like to underline such as: In the introduction part there are presented big paragraphs with no bibliography, From my point of view this aspect is worth improving.
Another aspect I would like to mention is the big number of figures and graphs, I belive some of them could be moved to the supporting info file.
Comments on the Quality of English Language
no comments
Reviewer 3 Report
Comments and Suggestions for Authors
1 Greetings to you. I have gone through your manuscript. Your study is very interesting and your innovative approach in bringing the mechanisms underlying endocrine resistance in luminal A breast cancer cells, particularly in response to tamoxifen, is highly commendable. I have compiled a set of questions across different dimensions of your study. I believe your responses to these questions will provide a more nuanced understanding of your methodology, addressing intricacies and potential areas for further discussion:
1. Could you provide a more nuanced elucidation on the specific lysosomal calcium channels, distinct from TRPML1 and TPCs, that might be implicated in the orchestration of tamoxifen-induced TFEB activation? Furthermore, how might the modulation of these channels impact the observed effects on cellular behavior and endocrine resistance?
2 2.Considering the reversible nature of TFEB nuclear translocation upon tamoxifen removal, I am curious about your insights into the potential long-term effects or lingering impacts on cellular behavior, particularly in tamoxifen-resistant cells. Do you anticipate any sustained alterations in cellular responses after the removal of tamoxifen, and if so, how might this influence subsequent therapeutic strategies?
3. In the intricate interplay between autophagy and drug resistance, could you expound upon how the increased nuclear accumulation of TFEB in tamoxifen-resistant cells influences the overall autophagic flux? Additionally, how does this altered autophagic flux contribute to the cells' ability to cope with stress, and are there specific molecular signatures associated with this phenomenon?
4. Given the central role of Ca2+ in the regulation of TFEB, I am interested in a more detailed discussion on potential cross-talk or interplay with other signaling pathways. How might these interconnected signaling networks contribute to or modulate the observed effects, and are there any specific feedback loops that warrant further exploration?
5. In your insightful discussion on the clinical relevance of lysosomal calcium channels as potential druggable targets, I am keen to understand the challenges and opportunities associated with targeting these channels. What are the specific considerations in translating these findings into novel therapeutic interventions, and are there potential off-target effects that merit careful consideration?
I look forward to your thoughtful responses.
Round 2
Reviewer 3 Report
Comments and Suggestions for Authors
The manuscript looks improved and is well-structured. Please take note of a few comments I have raised.
